# NOLA: COMPRESSING LoRA USING LINEAR COMBINATION OF RANDOM BASIS

Soroush Abbasi Koohpayegani [*,1]            K L Navaneet [*,1]

Parsa Nooralinejad[1]            Soheil Kolouri[2]            Hamed Pirsiavash[1]

[1]University of California, Davis            [2] Vanderbilt University

## ABSTRACT

Fine-tuning Large Language Models (LLMs) and storing them for each downstream task or domain is impractical because of the massive model size (e.g., 350GB in GPT-3). Current literature, such as LoRA, showcases the potential of low-rank modifications to the original weights of an LLM, enabling efficient adaptation and storage for task-specific models. These methods can reduce the number of parameters needed to fine-tune an LLM by several orders of magnitude. Yet, these methods face two primary limitations: (1) the parameter count is lower-bounded by the rank one decomposition, and (2) the extent of reduction is heavily influenced by both the model architecture and the chosen rank. We introduce NOLA, which overcomes the rank one lower bound present in LoRA. It achieves this by re-parameterizing the low-rank matrices in LoRA using linear combinations of randomly generated matrices (basis) and optimizing the linear mixture coefficients only. This approach allows us to decouple the number of trainable parameters from both the choice of rank and the network architecture. We present adaptation results using GPT-2, LLaMA-2, and ViT in natural language and computer vision tasks. NOLA performs as well as LoRA models with much fewer number of parameters compared to LoRA with rank one, the best compression LoRA can archive. Particularly, on LLaMA-2 70B, our method is almost 20 times more compact than the most compressed LoRA without degradation in accuracy. Our code is available here: https://github.com/UCDvision/NOLA

## 1 INTRODUCTION

Large pre-trained neural networks have exhibited remarkable generalization abilities across a diverse range of downstream tasks in both natural language processing and computer vision, achieving unprecedented data efficiency. For instance, large language models have demonstrated the capability for few-shot generalization (Brown et al., 2020) across a variety of tasks, including translation, question-answering, cloze tasks, and reasoning. Similarly, in DINOv2, (Oquab et al., 2023) showcase how a large pre-trained ViT model (Dosovitskiy et al., 2020) with more than 1B parameters yields superior all-purpose visual features for a variety of downstream benchmark tasks at both image and pixel levels. Typically, these pre-trained large models are adapted to downstream tasks through fine-tuning of their parameters. However, fine-tuning and storing the entire set of model parameters for each task incurs a significant storage cost (e.g., 350GB for GPT-3). This challenge has spurred a considerable body of recent works focusing on parameter-efficient fine-tuning of large models (Hu et al., 2021; Xu et al., 2023; Dettmers et al., 2023; Chen et al., 2022; Sung et al., 2022b).

Inspired by the low intrinsic dimensionality of over-parameterized networks' optimal parameters (Li et al., 2018; Aghajanyan et al., 2021), (Hu et al., 2021) proposed a seminal hypothesis that the change in weights during model adaptation/finetuning has a low "intrinsic rank", leading to the development of Low-Rank Adaptation (LoRA). In essence, LoRA enables the indirect training of a linear layer in a neural network by optimizing the rank-decomposition matrices for the weight change in these layers, resulting in a significant reduction in the number of parameters required for adaptation

---

* Equal Contribution.

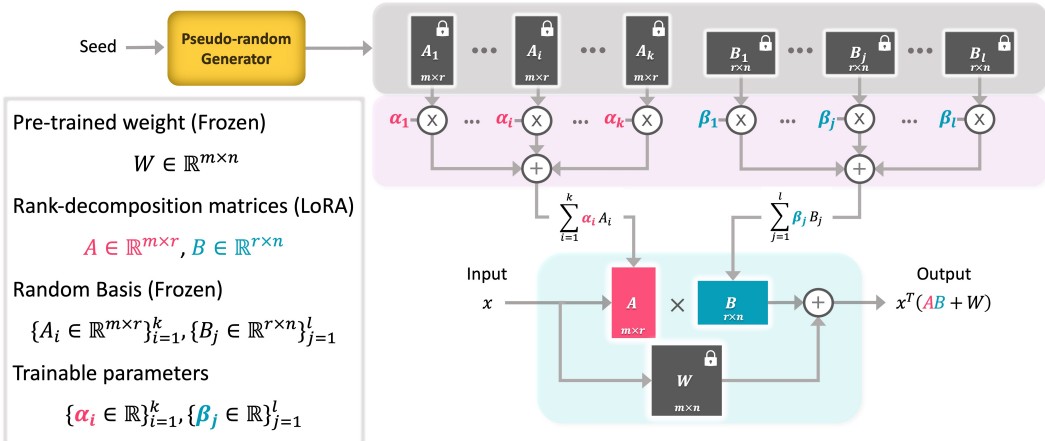

Figure 1: **Our Method (NOLA):** After constraining the rank of $\Delta W$ by decomposing it to $A \times B$, we reparametrize $A$ and $B$ to be a linear combination of several random basis matrices. We freeze the basis and $W$ and learn the combination coefficients. To reconstruct the model, we store the coefficients and the seed of the random generator which is a single scalar. NOLA results in more compression compared to LoRA and more importantly decouples the compression ratio from the rank and dimensions of $W$. One can reduce the number of parameters to 4 times smaller than rank=1 of LoRA which is not possible with LoRA due to rank being an integer number.

(e.g., 10,000× parameter reduction for GPT-3). Notably, LoRA has gained popularity, and various extensions of this method have been proposed since its inception (Xu et al., 2023; Dettmers et al., 2023). However, LoRA and its derivatives have three inherent limitations: (1) the parameter count is lower-bounded by the rank one decomposition of linear layers, and (2) the number of parameters is quantized since rank is an integer number, and (3) the number of parameters inherently depends on the model's architecture, i.e., the dimensions of the linear matrix, and the choice of rank. In this paper, we introduce a method, denoted as NOLA, that offers the same benefits as LoRA while addressing its limitations. NOLA allows one to decouple the number of trainable parameters from both the choice of rank and the network architecture, and it breaks the rank-one decomposition limit of LoRA.

NOLA is inspired by the recent work by Nooralinejad et al. (2022), titled PRANC. In this work, we reparameterize a neural network using a linear combination of pseudo-randomly generated weights. Then, we indirectly train the network parameters by optimizing the linear mixture coefficients. This approach results in a significant reduction in the total number of parameters needed to represent the network. Unlike PRANC, our focus in NOLA is on reparameterizing the change of neural weights for fine-tuning large models. More critically, unlike PRANC, NOLA incorporates the invaluable insight from (Hu et al., 2021), which posits that the weight change during fine-tuning is intrinsically low-rank. In essence, we utilize the rank-decomposition presented in LoRA but assemble the rank-decomposition matrices as a linear combination of pseudo-random matrices (i.e., the 'basis'). Optimizing the rank-decomposition matrices in NOLA is akin to determining the linear mixture coefficients for the random matrices. This design allows us to decouple the number of parameters from the shape of the linear layer and also from the rank choice. Furthermore, the low-rank constraints offer substantial advantages in compute and memory footprint over the methodology proposed in PRANC. Figure 1 illustrates the fundamental concept of NOLA.

**Why Fewer Parameters Matter?**

We envision a future where we must efficiently manage and transition between multiple Large Language Models (LLMs), each tailored for specific tasks. This vision arises from the necessity for LLMs customized with private data and/or the concept of crafting a universal LLM that can summon customized LLMs as a versatile toolbox to tackle diverse tasks (Schick et al., 2023). However, currently, customized LLMs demand substantial storage, and the process of switching between them lacks efficiency due to large I/O operations. NOLA offers a more compact reparameterization solution that can be stored effectively in GPU memory, allowing for on-demand reconstruction directly on the GPU itself when a new task arises.

Note that while storing parameters in CPU memory is a cost-effective option, the process of transferring them from CPU to GPU incurs substantial time and power consumption. Moreover, this

data transfer relies on a shared resource (e.g., PCIe bus), which may experience congestion in busy server environments. Therefore, optimizing model compactness to fit several of them within the limited GPU memory proves advantageous in these scenarios. As an example, 1,000 customized GPT-3 models using LoRA need almost 35GB of memory (assuming LoRA compresses it by a factor of $10,000\times$), which may not fit in the GPU memory along with the LLM model itself. Hence, compacting it by an additional factor of 5 reduces it to 7GB, which can fit in the GPU memory, leading to very efficient switching between the tasks.

**Contributions.** Our specific contributions in this paper are: 1) A novel reparameterization for compressing task-specific large language models, denoted as NOLA. 2) NOLA decouples the compression ratio from the rank and dimension of the weight matrix, unlocking higher compression ratios while keeping most benefits of LoRA, including reduced memory and computation at training time. 3) NOLA can be further improved by quantizing the coefficients and can be applied to other architectures like CNNs. 4) Applied to PRANC, NOLA speeds it up and reduces its memory footprint.

## 2 PROPOSED METHOD: NOLA

LoRA, short for Low-Rank Adaptation, is a widely embraced method for customizing a pre-trained model, such as GPT, for a specific task. Instead of changing all parameters denoted as $W$ within a given layer, LoRA maintains the original pre-trained parameters $W$ as a constant and learns a residual adjustment $\Delta W$ to fine-tune the model for the new task. The resulting updated layer parameters are then computed as $W + \Delta W$. The core concept behind LoRA is to minimize the size of $\Delta W$ by constraining its rank. In a more formal context, considering $W \in \mathbb{R}^{m \times n}$ and $\Delta W \in \mathbb{R}^{m \times n}$, LoRA accomplishes this by reparameterizing $\Delta W$ as the product of two matrices, $\Delta W = A \times B$, where $A \in \mathbb{R}^{m \times r}$ and $B \in \mathbb{R}^{r \times n}$, with $r$ representing the rank—a hyperparameter. By selecting a relatively small rank ($r << \min(m, n)$), LoRA efficiently reduces memory usage. This optimization is achieved by storing $A$ and $B$, which exhibit a significantly more compact representation than the full $\Delta W$. The resulting compression ratio is quantified as $\frac{mn}{r(m+n)}$. Unfortunately, this compression ratio is: (1) tied to the shape of the parameters $m$ and $n$, and hence the model architecture and (2) is upper-bounded by $\frac{mn}{m+n}$, i.e., for $r = 1$.

In this paper, we introduce a novel reparameterization technique for $\Delta W$ that effectively decouples the rank from the compression ratio, allowing for a compression ratio higher than $\frac{mn}{m+n}$, which corresponds to $r = 1$ in the LoRA framework. To achieve this, we draw inspiration from PRANC (Nooralinejad et al., 2022) and reparameterize matrices $A$ and $B$ to exist within a lower-dimensional space defined by a set of randomly generated basis matrices. Formally, we express this as:

$$A = \sum_{i=1}^{k} \alpha_i A_i \quad , \quad B = \sum_{j=1}^{l} \beta_j B_j \qquad (1)$$

where, $A_i \in \mathbb{R}^{m \times r}$ and $B_j \in \mathbb{R}^{r \times n}$ are random matrices generated by a Pseudo Random Number Generator with a fixed seed. We subsequently learn $A$ and $B$ as linear combinations of these predefined and frozen random matrices. Importantly, the random matrices themselves remain constant, and we optimize only the coefficient vectors $\alpha$ and $\beta$. Then:

$$\Delta W = \Big( \sum_{i=1}^{k} \alpha_i A_i \Big) \times \Big( \sum_{j=1}^{l} \beta_j B_j \Big) \qquad (2)$$

In practical terms, to store $\Delta W$ for a specific task, we only need to retain the seed (a single scalar) and the coefficient vectors $\alpha$ and $\beta$. Remarkably, this approach allows for a small number of basis matrices ($k + l$) to be chosen, irrespective of the rank of the $A \times B$ factorization and the shape of $\Delta W$, thereby enhancing the compression ratio to go beyond $\frac{mn}{m+n}$.

**Quantization of coefficients $\alpha$ and $\beta$:** We are mainly interested in reducing the storage for a new task, assuming the pre-trained LLM is already available. Hence, to further reduce the storage, we quantize the $\alpha$ and $\beta$ coefficients to lower precision (e.g., 4 bits) while the random basis and the pre-trained LLM weights have standard FP16 floating point precision. Note that one can also quantize $A$ and $B$ matrices in LoRA; however, our method does not force $A$ and $B$ themselves to be of low precision. One can quantize $\alpha$ and $\beta$ after the optimization (post-training quantization) or while optimizing them (quantization-aware training). We expect the latter to perform better. For quantization-aware learning, we use the method in (Rastegari et al., 2016b; Jacob et al., 2018) where we use the quantized $\alpha$ and $\beta$ in the forward pass and update the FP16 versions of $\alpha$ and $\beta$ in the

backward pass. Moreover, we use the Straight-Through Estimator (STE) trick (Bengio et al., 2013) to estimate the gradient.

A few recent works have shown that it is possible to quantize the weights of LLMs for each task, which reduces both computation and storage. However, these methods are not suitable for a large number of tasks since the quantized LLM is still task-specific and large.

**Memory Efficiency:** Note that depending on the number of basis matrices, the random basis may be large, requiring a large memory. Interestingly, generating random matrices in the GPU itself is very fast, so similar to PRANC, at each iteration, we generate chunks of the basis matrices at a time, multiply them by the corresponding coeficents, and discard them. Generating a basis on the fly at the inference time can drastically reduce the communication cost between CPU and GPU since $\alpha$ and $\beta$ vectors for several tasks can be stored in the GPU memory.

**Efficiency of NOLA compared to PRANC:** PRANC (Nooralinejad et al., 2022) reshapes the whole model parameters or each layer of it into a long vector and reparameterizes that by a linear combination of random vectors. However, as mentioned in (Nooralinejad et al., 2022), this method involves multiplication of the coefficients with the big random matrix twice at each iteration (once in forward and once in backward passes), which is very expensive. For instance, the size of the random matrix for ResNet18 with 1000 coefficients will be almost $11M \times 1K$. NOLA reduces this computation while keeping the same number of parameters by reshaping the long vector to be a 2D matrix and constraining its rank. Assuming $d^2$ weights and $k$ random basis, the basis matrix size for PRANC will be $kd^2$ while NOLA with rank $r$ reduces that to $kdr$ assuming that each component in $A \times B$ has $\frac{k}{2}$ basis matrices to keep the number of parameters equal to PRANC. Then, the total compute for PRANC will be $kd^2 + d^2 \approx kd^2$ while for NOLA, it will be $kdr + 2dr \approx kdr$. Hence, assuming a small rank $r$, NOLA can reduce the training time of PRANC by a large factor $\frac{d}{r}$ due to the reduction of the computation at forward and backward passes. Moreover, in the appendix, we empirically show that NOLA offers a better coverage of the weight space compared to PRANC.

**Structure of the parameters:** Note that one can apply NOLA to model architectures other than transformer by simply reshaping the weight tensor to be a 2D matrix (preferably close to square) and then compressing it. We do this in our ResNet experiments in the Appendix, where the weight matrices are 4D tensors of convolutional filters.

## 3 EXPERIMENTS

Here, we evaluate NOLA in transfer learning tasks in both NLP and vision. Moreover, in the appendix, we evaluate NOLA in training from scratch.

### 3.1 NOLA ON GPT-2:

We adapt the parameters of pre-trained GPT-2 to three different Natural Language Generation (NLG) datasets by finetuning the parameters using NOLA. We use GPT-2-Large and GPT-2-Medium in our experiments. We follow the (Li & Liang, 2021; Hu et al., 2021) for our adaptation setup.

**Datasets:** We utilize the following datasets for our Natural Language Generation (NLG) task: E2E NLG Challenge (Novikova et al., 2017) serves as a commonly used benchmark for evaluating NLG models. It encompasses of 51,200 samples, distributed as follows: 42,200 for training, 4,600 for validation, and an additional 4,600 for testing. DART (Nan et al., 2020) is yet another significant dataset employed for evaluating text-to-data generation. This dataset is rich with 82,191 examples drawn from various domains. WebNLG (Gardent et al., 2017) is a text-to-data dataset, boasting 22,000 examples spanning 14 distinct categories. Notably, the WebNLG test set introduces five new categories, prompting us to present results across all categories within this dataset. These datasets collectively provide a comprehensive foundation for our NLG evaluation and experimentation.

**LoRA:** In our experiments, we apply LoRA on both query and value projection layer in each attention block. Since number of parameters is tied to the rank, we adjust the rank to reduce number of parameters. We compare to LoRA with both rank four and rank one.

**Other Baselines:** Moreover, we compared NOLA to a few other baselines, including finetuning all parameters, Adapters (Houlsby et al., 2019; Lin et al., 2020b; Pfeiffer et al., 2021; Rücklé et al., 2020), and Prefix-layer tuning (PreLayer) (Li & Liang, 2021).

Table 1: **E2E NLG Challenge:** We compare NOLA to LoRA with two different architectures: GPT-2 medium (M) and large (L). QV refers to Query and Value projection, while MLP denotes the MLP layer within transformers. Adapter[L] and Adapter[H] are two Adapter baselines reported in (Hu et al., 2021). The best and second best performing methods are in bold. To reduce number of parameters in LoRA, we use lower rank (LoRA rank=1). We don't see drop in performance by reducing number of trainable parameters to $\frac{1}{20}$ of LoRA with rank 4 in GPT-L. Note that in LoRA, one cannot reduce the number of parameters below rank one.

| **GPT-2 M** | | | | | | | | |
|---|---|---|---|---|---|---|---|---|
| Method | Adapted Layers | Adapter Rank | # Trainable Parameters | E2E NLG Challenge | | | | |
| | | | | BLEU | NIST | MET | ROUGE-L | CIDEr |
| Finetune | All Layers | - | 354.920M | 68.2 | 8.62 | 46.2 | 71.0 | 2.47 |
| Adapter[L] | Extra Layers | - | 0.370M | 66.3 | 8.41 | 45.0 | 69.8 | 2.40 |
| Adapter[L] | Extra Layers | - | 11.090M | 68.9 | 8.71 | 46.1 | 71.3 | 2.47 |
| Adapter[H] | Extra Layers | - | 11.090M | 67.3 | 8.50 | 46.0 | 70.7 | 2.44 |
| Finetune[Top2] | Last 2 Layers | - | 25.190M | 68.1 | 8.59 | 46.0 | 70.8 | 2.41 |
| PreLayer | Extra Tokens | - | 0.350M | 69.7 | 8.81 | 46.1 | 71.4 | 2.49 |
| LoRA | QV | 4 | 0.350M | **70.4** | **8.85** | **46.8** | **71.8** | **2.53** |
| LoRA | QV | 1 | 0.098M | 68.7 | 8.72 | 45.6 | 70.5 | 2.43 |
| NOLA (Ours) | QV | 8 | 0.350M | 70.1 | 8.80 | **46.8** | 71.7 | **2.53** |
| NOLA (Ours) | QV | 8 | 0.096M | 70.0 | **8.82** | 46.7 | 71.6 | 2.51 |
| NOLA (Ours) | MLP | 8 | 0.096M | 70.2 | 8.79 | 46.7 | **71.8** | 2.51 |
| NOLA (Ours) | QV | 8 | 0.048M | 70.1 | **8.82** | 46.4 | 71.4 | 2.52 |
| NOLA (Ours) | MLP | 8 | 0.048M | 69.4 | 8.71 | 46.5 | 71.5 | 2.47 |
| **GPT-2 L** | | | | | | | | |
| Finetune | All Layers | - | 774.030M | 68.5 | 8.78 | 46.0 | 69.9 | 2.45 |
| Adapter[L] | Extra Layers | - | 0.880M | 69.1 | 8.68 | 46.3 | 71.4 | 2.49 |
| Adapter[L] | Extra Layers | - | 23.000M | 68.9 | 8.70 | 46.1 | 71.3 | 2.45 |
| PreLayer | Extra Tokens | - | 0.770M | 70.3 | **8.85** | 46.2 | **71.7** | 2.47 |
| LoRA | QV | 4 | 0.770M | **70.4** | **8.89** | **46.8** | **72.0** | 2.47 |
| LoRA | QV | 1 | 0.184M | 69.9 | 8.81 | 46.7 | 71.6 | **2.53** |
| NOLA (Ours) | QV | 8 | 0.144M | **70.5** | **8.85** | **46.8** | **71.7** | **2.54** |
| NOLA (Ours) | MLP | 8 | 0.144M | 70.1 | 8.80 | 46.5 | 71.2 | 2.52 |
| NOLA (Ours) | QV | 8 | 0.072M | 69.8 | 8.80 | 46.4 | 71.3 | 2.51 |
| NOLA (Ours) | MLP | 8 | 0.072M | 69.4 | 8.71 | 46.6 | 71.5 | 2.48 |
| NOLA (Ours) | QV | 8 | 0.036M | 70.1 | 8.80 | 46.7 | 71.7 | **2.53** |
| NOLA (Ours) | MLP | 8 | 0.036M | 70.0 | 8.81 | 46.4 | 71.5 | **2.53** |

**NOLA:** We evaluate NOLA with two different variations: 1. Adapting MLP layers. 2. Adapting query and value projection matrices. Note that, unlike LoRA, we can use any number of parameters while applying NOLA to any weight structure since the number of parameters is not tied to the shape of the weight tensor. We allocate an equal number of parameters to $A$ and $B$ in each NOLA layer (i.e., $k = l$). Using $k = l = 1000$ results in $0.096M$ parameters in GPT-M and $0.144M$ parameters in GPT-L. Also, we use half ($k = l = 500$) and quarter ($k = l = 250$) number of parameters per layer to get smaller checkpoints.

**Implementation Details:** We trained our models using a single NVIDIA RTX 6000 Ada Generation GPU. For all hyperparameters except learning rate, we use the same values as LoRA for training and evaluation of GPT-2. We train our models for 5 epochs with a learning rate of 0.1 and no weight decay. We use a batch size of 8. We use a rank of 8 for NOLA in our experiments. Like LoRA, we scale $A \times B$ with $\frac{c}{r}$, where $c$ is a hyperparameter and $r$ is the rank. We use the default value of $c = 1$.

**Results:** We compare to LoRA and other baselines in Table 1 and Table 11 in the Appendix. NOLA is on par or better compared to other methods with the same number of parameters. In the E2E task, NOLA with $0.036M$ parameters archives a BLEU score of $70.12$, which is $20$ times more compact compared to LoRA with rank $4$ that has $0.77M$ parameters and archives a BLEU score of $70.4$. This NOLA model uses a rank of $8$, which does not affect the number of parameters and increases the run time slightly (negligible compared to that of the actual LLM model). Note that our goal is to reduce the number of parameters without reducing the accuracy, which is shown in Tables 1 and 11. We are not claiming that NOLA improves the accuracy compared to baselines. We show various variants of NOLA (MLP, QV, etc) to emphasize that NOLA is not very sensitive to the choice of the variation.

**Training Time and Memory of NOLA:** Similar to LoRA, in the inference time, we can calculate $A \times B$ offline and merge it with $W$. Therefore, NOLA does not have any overhead compared to the original model. In training time, NOLA has a small overhead due to the multiplication of coefficients

Table 2: **Training time and memory:** We compare the training memory and running time of NOLA to LoRA. Since generating a random basis on each layer has a small overhead, we can generate random basis once and share the basis across layers to save time. This version of NOLA has a similar runtime to LoRA and on-par accuracy to NOLA with a non-shared basis.

| Model & Method | Random Basis | Training Memory | Training Time (ms/batch) | # Trainable Parameters | E2E NLG Challenge | | | | |
|---|---|---|---|---|---|---|---|---|---|
| | | | | | BLEU | NIST | MET | ROUGE-L | CIDEr |
| GPT-2 L (LoRA r=1) | - | 33.35GB | 776 | 184K | 69.89 | 8.81 | 46.70 | 71.64 | 2.53 |
| GPT-2 L (NOLA QV) | Non-shared | 33.37GB | 834 | 144K | 70.46 | 8.85 | 46.77 | 71.68 | 2.54 |
| GPT-2 L (NOLA QV) | Shared | 33.40GB | 779 | 144K | 70.32 | 8.85 | 46.74 | 71.71 | 2.54 |

Table 3: **Effect of rank in NOLA:** We vary the rank from 1 to 8. Note that we can use the same number of parameters in all ranks since the number of parameters is not tied to the rank in NOLA.

| # Train Params | Rank | E2E NLG Challenge | | | | |
|---|---|---|---|---|---|---|
| | | BLEU | NIST | MET | ROUGE-L | CIDEr |
| GPT-2 M (NOLA QV) | | | | | | |
| 96K | 8 | 70.03 | 8.82 | 46.74 | 71.64 | 2.51 |
| 96K | 4 | 69.69 | 8.76 | 46.56 | 71.44 | 2.51 |
| 96K | 2 | 70.47 | 8.86 | 46.71 | 71.79 | 2.53 |
| 96K | 1 | 69.09 | 8.78 | 45.87 | 70.15 | 2.45 |
| 96K | 8 | 70.03 | 8.82 | 46.74 | 71.64 | 2.51 |
| 48K | 8 | 70.09 | 8.82 | 46.44 | 71.36 | 2.52 |
| 24K | 8 | 68.30 | 8.67 | 45.13 | 68.91 | 2.40 |
| 12K | 8 | 67.18 | 8.60 | 43.99 | 68.38 | 2.26 |
| GPT-2 L (NOLA QV) | | | | | | |
| 144K | 8 | 70.46 | 8.85 | 46.77 | 71.68 | 2.54 |
| 144K | 4 | 70.25 | 8.85 | 46.84 | 71.81 | 2.54 |
| 144K | 2 | 69.69 | 8.78 | 46.55 | 71.25 | 2.51 |
| 144K | 1 | 69.71 | 8.82 | 46.47 | 70.96 | 2.51 |

Table 4: **Quantization of coefficients:** Post-training quantization of parameters does not degrade the performance up to the 4 bit quantization. In quantization-aware training, NOLA is more robust to quantization compared to LoRA.

| Model & Method | # Quant Bits | E2E NLG Challenge | | | | |
|---|---|---|---|---|---|---|
| | | BLEU | NIST | MET | ROUGE-L | CIDEr |
| Post Training Quantization | | | | | | |
| GPT-2 L (LoRA r=1) | 16-bit | 69.89 | 8.81 | 46.70 | 71.64 | 2.53 |
| | 8-bit | 69.91 | 8.81 | 46.69 | 71.75 | 2.53 |
| | 4-bit | 69.63 | 8.75 | 46.32 | 71.24 | 2.48 |
| | 3-bit | 62.01 | 8.02 | 42.01 | 67.23 | 2.07 |
| GPT-2 L (NOLA QV) | 16-bit | 70.46 | 8.85 | 46.77 | 71.68 | 2.54 |
| | 8-bit | 70.43 | 8.84 | 46.78 | 71.72 | 2.54 |
| | 4-bit | 70.29 | 8.82 | 46.74 | 71.82 | 2.52 |
| | 3-bit | 65.14 | 8.58 | 44.38 | 67.56 | 2.23 |
| Quantization Aware Training | | | | | | |
| GPT-2 L (LoRA r=1) | 3-bit | 67.08 | 8.86 | 44.67 | 68.76 | 2.36 |
| | 2-bit | 56.13 | 4.70 | 35.38 | 63.68 | 1.40 |
| GPT-2 L (NOLA QV) | 3-bit | 70.14 | 8.82 | 46.58 | 71.61 | 2.53 |
| | 2-bit | 68.69 | 8.72 | 46.06 | 70.61 | 2.48 |

to basis weights. We measure the running time and memory footprint of NOLA during training and compare it to LoRA in Table 2. Since generating a random basis for each layer adds a small overhead, we can share the random basis across all layers and generate and store them only once to improve the running time. We measure time and memory with a batch size of 8. NOLA, with a unique random basis for each layer, is slightly slower than LoRA. However, NOLA with a shared random basis has on-par accuracy with the unique random basis and has a similar running time to LoRA.

**Ablation Study on the rank of NOLA:** Since the number of parameters is decoupled from the rank of the matrix, we can solely evaluate the effect of rank without changing the number of parameters. We report the effect of rank in Table 3. We vary the rank from 1 to 8 and use $c = 1.0$ for ranks 4 and 8, and $c = 0.5$ for lower ranks. As also noted by (Hu et al., 2021), understanding the effect of rank needs more rigorous study as future work.

## 3.2 NOLA WITH QUANTIZED COEFFICIENTS, $\alpha$ AND $\beta$:

We evaluate the performance of NOLA with rank 4 with quantized $\alpha$ and $\beta$ parameters on the E2E dataset in Table 4 in two different setups. First, we do Post Training Quantization (PTQ) by quantizing the parameters to $q$ bits after the training. We observe no significant drop in both LoRA and NOLA in 4 bits PTQ experiments. Second, we evaluate models with Quantization Aware Training (QAT) where we quantize the coefficients in the forward pass and update them in the non-quantized form. In QAT with 3 bits, NOLA has a slight drop of 0.3 points while LoRA has a drop of 2.8 points in the BLEU metric. Note that in NOLA, although $\alpha$ and $\beta$ are quantized, $A$ and $B$ in Eq 1 are not quantized since the basis matrices, $A_i$ and $B_j$, are non-quantized. This is in contrast to LoRA where $A$ and $B$ are quantized.

## 3.3 NOLA ON LLAMA-2:

Finetuning with LoRA for larger LLMs is challenging due to compute demand, so we need to resort to QLoRA where the base model is quantized. Our NOLA framework is agnostic to the quantization of base model, so for LLaMA-2 (Touvron et al., 2023) experiments, we use NOLA with base model quantized to 8-bits while the NOLA coefficients still use 16-bit. We fine-tune the pretrained LLaMA-2

Table 5: **Instruction finetuning for quantized LLaMA-2:** NOLA fine-tunes LLaMA-2 70B (8-bit) with $0.57M$ parameters, a $95\%$ reduction compared to rank-one LoRA. We use quantized version of LLaMA-2 for both LoRA and NOLA, so our LoRA baseline is equivalent to QLoRA.

| | LLaMA-2 - 7B (8-bit) | | | LLaMA-2 - 13B (8-bit) | | | LLaMA-2 - 70B (8-bit) | | |
| | w/o Finetuning | LoRA | NOLA | w/o Finetuning | LoRA | NOLA | w/o Finetuning | LoRA | NOLA |
|---|---|---|---|---|---|---|---|---|---|
| Adapter Rank | - | 1 | 16 | - | 1 | 16 | - | 1 | 16 |
| Trainable Parameters | - | 2.50M | 0.06M ($\downarrow 97\%$) | - | 3.91M | 0.14M ($\downarrow 96\%$) | - | 12.94M | 0.57M ($\downarrow 95\%$) |
| Train Loss | 1.53 | 0.97 | 1.05 | 1.43 | 0.94 | 0.95 | 1.42 | 0.87 | 0.90 |
| Val Loss | 1.74 | 1.04 | 1.01 | 1.59 | 0.96 | 0.97 | 1.53 | 0.92 | 0.90 |
| MMLU Acc | 45.3 | 46.5 | 46.5 | 54.8 | 55.3 | 55.3 | 68.9 | 69.5 | 69.4 |

model using 8-bit QLoRA on the Alpaca dataset (Taori et al., 2023), reporting both training and validation loss metrics specific to the Alpaca dataset. Additionally, we employ the MMLU (Massively Multitask Language Understanding) benchmark (Hendrycks et al., 2021) to assess performance across a diverse set of language understanding tasks. This benchmark comprises 57 tasks spanning areas such as mathematics, history, computer science, and law. On MMLU, we focus on 5-shot evaluation (providing 5 examples in the context), following the standard practice.

**Implementation Details:** We apply LoRA and NOLA across all layers (Query, Key, Value and Output projections and MLP layers) of the transformer on three different model sizes of LLaMA-2: 7B, 13B, and 70B. For our LoRA experiments, we set the rank $r = 1$ since LoRA paper (its Table 15) shows that rank one model performs as well as higher ranks for larger LLMs (e.g., GPT-3). For NOLA, we use $r = 16$ and adjust the number of optimized parameters for each LLaMA-2 model size using the following settings: $k = l = 128$ for LLaMA-2 7B, $k = l = 256$ for LLaMA-2 13B, and $k = l = 512$ for LLaMA-2 70B. During fine-tuning LoRA or NOLA, we adhere to the hyperparameters reported in QLoRA, (Dettmers et al., 2023). We optimize for one epoch on the Alpaca dataset with a batch size of 256 using four RTX 3090 GPUs. The learning rate is 0.0002 for LoRA and 0.001 for NOLA. Similar to LoRA, we scale $A \times B$ with $\frac{c}{r}$, where $c$ is a hyperparameter and $r$ is the rank. We use $c = 16$ for LoRA and $c = 4$ for NOLA.

**Results:** Results are presented in Table 5. Remarkably, NOLA is capable of fine-tuning LLaMA-2 70B with fewer than $0.6M$ parameters, representing an average reduction of parameters by $95\%$ compared to LoRA with rank one.

## 3.4 NOLA ON VISION TRANSFORMERS

We perform experiments on the image classification task on ViT-B and ViT-L architectures with both supervised and self-supervised (MAE) initializations.

**Implementation details:** All pre-trained networks are obtained from Timm library (Whitman). All approaches are trained for 50 epochs, and the top-1 accuracy at the final epoch is reported. We use a batch-size of 64 and tune the initial learning rate for each dataset and architecture for all approaches. Since our focus is on finetuning on small datasets, we use 5 and 10 labeled examples per class for finetuning. Since there is a high variance in performance due to the small training sets, we sample four different sets of training samples per $k$-shot and three different initializations for LoRA/NOLA and report the mean accuracy and standard deviation. All approaches are trained with cross-entropy loss. Additional details are in the appendix.

**Datasets:** ImageNet-21k and ImageNet-1k are used to pretrain the backbone models. We use CIFAR10 (Krizhevsky et al., 2014), CIFAR100 (Krizhevsky et al., 2009), CUB-200-2011 (Welinder et al., 2010), Caltech-101 (Fei-Fei et al., 2004), Aircraft (Maji et al., 2013), Food101 (Bossard et al., 2014), Pets (Parkhi et al., 2012) and SUN397 (Xiao et al., 2010) datasets for finetuning.

**Baselines:** We compare NOLA with three baseline approaches: Linear, Full-FT (full fine-tuning) and LoRA (Hu et al., 2021). In Linear, only the final classifier head is optimized, while in Full-FT, the entire backbone network is optimized too. No additional parameters are used during finetuning for either of these approaches. We apply LoRA on Query, Key, and Value projection matrices with rank set 4 for ViT-B to 1 or 4 for ViT-L. In our preliminary experiments, LoRA with rank one sees a big drop in accuracy. This is aligned with our GPT-2 M experiments where smaller models require higher rank. We apply NOLA on the MLP layers and use rank of 4 for ViT-B and 1 for ViT-L. We report the number of trainable parameters for each approach excluding the classifier head parameter count which is common to all approaches. We also report nearest-neighbor (1-NN) evaluation for zero-shot classification on downstream tasks using ImageNet pretrained features.

Table 6: **Results on vision transformers.** We finetune ImageNet pre-trained ViT models on multiple small datasets with 5 and 10 training samples. The mean accuracy and standard deviation across 12 runs are reported. The number of train parameters for Linear classifier depends on the number of classes in the dataset (0.01, 0.1, 0.2, 0.1M parameters for CIFAR-10, CIFAR-100, CUB and Caltech respectively). We do not count the linear layer in trainable parameters. The best performing method is in bold while all methods within one point of the best are underlined. NOLA outperforms LoRA with comparable parameters across datasets and architectures, particularly in the low training data regime. The performance of NOLA with half or one-third of the training parameters is comparable to that of LoRA. Note that LoRA with rank one is the most compact LoRA.

| Base Model | | # Train Params | CIFAR-10 5 | CIFAR-10 10 | CIFAR-100 5 | CIFAR-100 10 | CUB-200-2011 5 | CUB-200-2011 10 | Caltech-101 5 | Caltech-101 10 |
|---|---|---|---|---|---|---|---|---|---|---|
| ViT-B | Nearest Neighbor | | 79.6 | 80.8 | 52.4 | 59.2 | 71.9 | 78.0 | 84.1 | 87.5 |
| | Linear | 0 | 80.8 (1.1) | 85.1 (1.0) | 58.9 (0.9) | 64.5 (0.7) | 72.7 (0.4) | 79.2 (0.2) | 85.8 (0.8) | 88.5 (0.4) |
| | Full-FT | 5.3M | 73.9 (6.5) | 87.6 (2.7) | 61.4 (2.4) | 78.2 (1.1) | 59.7 (1.9) | 76.6 (0.2) | 87.9 (0.8) | 91.1 (0.5) |
| | LoRA (r=4) | 141K | 87.3 (2.3) | 93.1 (0.5) | 76.3 (0.5) | 81.6 (0.9) | 75.7 (0.5) | 82.4 (0.3) | 88.4 (1.1) | 90.8 (0.5) |
| | NOLA-MLP | 47K | 87.9 (1.3) | 92.2 (0.5) | 75.1 (0.6) | 81.3 (0.8) | 75.5 (0.6) | 81.7 (0.4) | 88.0 (1.2) | 90.6 (0.5) |
| ViT-B-MAE | Nearest Neighbor | | 18.2 | 19.8 | 5.8 | 9.8 | 13.2 | 25.3 | 28.2 | 40.7 |
| | Linear | 0 | 27.4 (1.9) | 34.5 (1.4) | 15.7 (0.7) | 22.2 (0.2) | 12.7 (0.3) | 18.4 (0.3) | 66.9 (1.1) | 76.9 (0.6) |
| | Full-FT | 5.3M | 41.1 (4.4) | 58.4 (3.6) | 19.7 (4.8) | 24.2 (11.1) | 23.0 (3.8) | 51.9 (2.8) | 76.4 (2.3) | 86.5 (0.5) |
| | LoRA (r=4) | 141K | 54.7 (1.6) | 70.1 (2.2) | 39.3 (3.1) | 52.4 (1.3) | 35.7 (1.5) | 54.0 (0.6) | 82.4 (0.6) | 87.7 (0.5) |
| | NOLA-MLP | 47K | 55.1 (2.6) | 72.1 (2.7) | 42.1 (1.4) | 53.5 (1.0) | 35.8 (1.5) | 53.9 (0.9) | 88.0 (1.2) | 90.6 (0.5) |
| ViT-L | Nearest Neighbor | | 88.7 | 89.9 | 68.9 | 74.0 | 77.4 | 82.3 | 88.4 | 90.1 |
| | Linear | 0 | 84.1 (1.8) | 88.4 (1.1) | 63.7 (1.3) | 70.6 (0.9) | 73.7 (0.6) | 79.2 (0.3) | 87.6 (0.9) | 89.9 (0.4) |
| | Full-FT | 289M | 77.2 (2.7) | 90.2 (2.8) | 74.0 (2.3) | 86.2 (0.6) | 73.3 (0.9) | 83.9 (0.2) | 88.7 (1.0) | 91.3 (0.7) |
| | LoRA (r=4) | 375K | 86.5 (2.0) | 93.8 (1.0) | 82.9 (0.9) | 87.6 (0.6) | 81.2 (0.4) | 85.3 (0.3) | 89.3 (0.7) | 91.3 (0.3) |
| | LoRA (r=1) | 94K | 86.3 (1.3) | 92.8 (0.8) | 82.2 (0.8) | 85.6 (0.9) | 80.6 (0.3) | 85.2 (0.3) | 89.9 (1.0) | 91.6 (0.4) |
| | NOLA-MLP | 94K | 89.0 (3.6) | 96.0 (0.5) | 83.6 (0.9) | 87.8 (0.6) | 80.8 (0.6) | 85.2 (0.2) | 90.0 (0.7) | 91.7 (0.3) |
| | NOLA-MLP | 47K | 83.9 (1.8) | 93.0 (1.7) | 81.2 (1.0) | 87.1 (0.6) | 80.7 (0.5) | 85.0 (0.3) | 89.8 (0.8) | 91.5 (0.4) |
| ViT-L-MAE | Nearest Neighbor | | 33.5 | 39.2 | 15.2 | 21.9 | 16.9 | 29.2 | 57.4 | 67.6 |
| | Linear | 0 | 40.2 (2.3) | 49.2 (2.6) | 22.6 (0.9) | 31.3 (0.5) | 15.2 (0.3) | 21.9 (0.4) | 75.2 (0.5) | 83.2 (0.6) |
| | Full-FT | 289M | 60.6 (4.5) | 68.3 (4.0) | 37.9 (11.1) | 52.0 (16.1) | 42.2 (2.3) | 67.1 (1.1) | 87.2 (0.8) | 90.8 (0.7) |
| | LoRA (r=4) | 375K | 63.5 (3.0) | 82.4 (2.3) | 50.2 (6.8) | 62.6 (5.2) | 35.2 (2.9) | 60.8 (1.2) | 87.0 (0.9) | 90.7 (0.4) |
| | LoRA (r=1) | 94K | 67.7 (3.8) | 83.8 (1.2) | 50.4 (1.0) | 62.5 (0.6) | 32.9 (1.8) | 56.6 (1.7) | 87.0 (0.6) | 90.8 (0.4) |
| | NOLA-MLP | 94K | 70.6 (3.8) | 86.0 (1.4) | 51.7 (1.1) | 63.8 (0.8) | 36.9 (5.6) | 61.6 (1.0) | 87.4 (0.4) | 90.9 (0.5) |
| | NOLA-MLP | 47K | 69.6 (3.8) | 84.8 (1.1) | 49.9 (0.8) | 62.8 (0.7) | 36.1 (0.8) | 58.8 (1.2) | 87.1 (0.6) | 90.9 (0.4) |

Table 7: **More results on vision tasks.** Using ViT-B, NOLA achieves comparable performance as LoRA (r=4) with just one-third the number of parameters on four challenging datasets. The linear layer sizes are: 0.03M, 0.2M, 0.1M, 0.3M for Aircraft, Food101, Pets and SUN397 respectively.

| Method | # Train Params | Aircraft 5 Shot | Aircraft 10 Shot | Food101 5 Shot | Food101 10 Shot | Pets 5 Shot | Pets 10 Shot | SUN397 5 Shot | SUN397 10 Shot |
|---|---|---|---|---|---|---|---|---|---|
| Nearest Neighbor | | 24.6 | 27.1 | 48.4 | 54.2 | 82.3 | 86.2 | 44.4 | 51.5 |
| Linear | 0 | 29.7 (2.3) | 36.9 (2.1) | 53.2 (0.3) | 61.5 (0.1) | 88.4 (0.4) | 91.3 (0.3) | 38.0 (0.8) | 42.7 (0.4) |
| Full-FT | 82M | 31.4 (2.4) | 43.2 (2.0) | 48.6 (5.1) | 65.8 (2.7) | 82.7 (1.1) | 91.1 (0.5) | 45.0 (3.3) | 52.6 (0.3) |
| LoRA (r=4) | 0.141M | 32.4 (1.4) | 43.8 (1.5) | 60.8 (1.6) | 73.1 (0.6) | 85.5 (0.8) | 91.6 (0.5) | 51.6 (0.4) | 55.6 (0.3) |
| NOLA-MLP | 0.047M | 33.7 (2.2) | 43.3 (1.4) | 64.5 (0.8) | 72.6 (0.4) | 88.0 (0.6) | 92.2 (0.3) | 50.5 (0.4) | 55.5 (0.3) |

**Results:** Results on finetuning on image classification tasks are presented in Table 6. A naive Nearest Neighbor approach performs competitively on the CUB and Caltech datasets. LoRA and NOLA are significantly better than Linear and Full-FT on several settings (e.g. CIFAR-100 5 shot on ViT-B-MAE). This might be due to the overfitting of the Linear and Full-FT approaches on the limited number of train samples. When using a similar number of training parameters, NOLA outperforms LoRA in most of the settings across architectures and datasets. It also achieves comparable performance to LoRA with just half or one-third of the training parameters of LoRA. This is consistent with our observations on the NLG tasks. The difference between the two methods is particularly noticeable when the number of training examples is small - either in 5 shot setup or when the number of classes is small, as in CIFAR-10. This suggests that the improvement obtained by NOLA could be due to the reduction in the number of training parameters. Both LoRA and NOLA consistently and significantly outperform Linear and Full-FT approaches. NOLA can easily be employed in MLP layers since the number of training parameters is decoupled from the weight matrix dimensions. A similar application of LoRA would require $8\times$ more training parameters due to the large hidden layer dimensionality of the MLP module. We empirically observe that NOLA-MLP slightly outperforms NOLA on attention block (see Table 10 in appendix). We provide results on four additional datasets used for benchmarking transfer learning in Table 7. Aircraft, Food101 and Pets are finegrained datasets while SUN397 is a large dataset with 397 classes. There is a bigger difference in the performance of Nearest Neighbor and Linear approaches on most of these datasets

compared to those in Table 6, suggesting that it is harder to adapt to these datasets. In line with our prior observations in Table 6, NOLA with just one-third the number of parameters performs comparably to LoRA.

## 4 RELATED WORKS

**Vision and Language Transformer Models:** Transformer networks, introduced by (Vaswani et al., 2017), emerged as a sequence-to-sequence model in Natural Language Processing (NLP). Their success soon extended to the computer vision community, with subsequent works (Dosovitskiy et al., 2021; Touvron et al., 2021) introducing the Vision Transformer (ViT) network as a strong alternative to the Convolutional Neural Network (CNN) backbones. Transformers accept a sequence of tokens as input. These tokens can be, for instance, word embeddings in language or image patches in vision. BERT (Devlin et al., 2019) and GPT-2 (Radford et al., 2018) in NLP, and MAE (He et al., 2021) and DINO (Caron et al., 2021) in computer vision train transformer networks via self-supervision on large amounts of unlabeled data. These studies demonstrate that large transformer networks when trained on massive corpora, generalize well to downstream tasks even when finetuning on very few task-specific examples. For example, (Brown et al., 2020) show that GPT-3 with 175B parameters is a good few-shot learner. Lastly, the scaling law presented by (Kaplan et al., 2020) indicates that a simultaneous increase in training data and model parameters can lead to significant performance gains and emergent capabilities previously unavailable to smaller models.

**Parameter Efficient Fine-Tuning:** Owing to their unprecedented few-shot generalization performance, large neural networks, such as foundation models and LLMs have gained immense popularity in recent years. An increasing number of users are customizing these models to adapt them to their specific tasks. However, given the colossal size of these models, fine-tuning and storing the entire set of model parameters (Devlin et al., 2019; Radford et al., 2018) for each task is impractical. This challenge is exacerbated as the number of tasks increases. In addition to storage concerns, the overhead involved in loading task-specific models and transferring weights from CPU to GPU often becomes a computational bottleneck in many applications. Parameter Efficient Fine-Tuning (PEFT) approaches aim to address these issues by identifying the minimum number of parameters needed to adapt a large model. Adapters (Houlsby et al., 2019; Rebuffi et al., 2017; Lin et al., 2020b; Mahabadi et al., 2021) are PEFT approaches that achieve adaptation by adding small modules to the intermediate layers of the model. A major drawback of Adapters is the extra latency they introduce in inference. BitFit (Zaken et al., 2021) only adapt bias of the network. Ladder tuning (Sung et al., 2022a) reduce memory footprint in training by avoiding back-propagation through the main backbone. IA3 (Liu et al., 2022) trains extra parameters in the attention module. Another widely adopted PEFT approach is prompt-tuning for LLMs that involves optimizing a new set of input tokens, or prompts, for each task (Li & Liang, 2021; Lester et al., 2021; Hambardzumyan et al., 2020; Liu et al., 2021). While reminiscent of prompt engineering, the distinction lies in training a specific set of prompt tokens in prompt-tuning which might also increase inference latency.

(Hu et al., 2021) introduced LoRA, demonstrating that a low-rank modification of the original weights is sufficient to adapt an LLM to a new task. Unlike adapters and prompt-tuning, these low-rank modifications can be integrated into the original weights, thus avoiding additional overhead during inference. However, LoRA has two main limitations: 1) the rank-one decomposition sets a lower bound on the parameters needed for fine-tuning, and 2) the number of required parameters is contingent upon the architecture and the rank choice. Our work, termed NOLA, addresses these challenges by decoupling the trainable parameters from both the rank choice and the network architecture. Several recent studies have sought to enhance LoRA by quantizing its parameters (Dettmers et al., 2023; Xu et al., 2023; Kwon et al., 2022; Gong et al., 2023), optimizing the design choice of LoRA through neural architecture search (Zhang et al., 2022), or dynamically allocating parameters based on the required rank for each weight matrix (Zhang et al., 2023). Most of these enhancements are also compatible with our proposed method. In fact, we demonstrate that NOLA can be quantized to 4-bit without any performance degradation, thereby emphasizing that the concept of quantization is distinct from and complementary to, NOLA.

**Compact Deep Learning Models:** A closely related area to PEFT is model compression. Pruning (Kim et al., 2020; Lin et al., 2020a; Siems et al., 2021; Tiwari et al., 2021; Hayou et al., 2020; Wang et al., 2020; Li et al., 2021) and quantization (Rastegari et al., 2016a; Lee et al., 2021) stand as the principal methods for compressing neural networks. Techniques such as those in (Kusupati et al., 2020; Isik et al., 2022) can achieve a high pruning rate, leading to significant compression.

**Acknowledgment:** This work was partially supported by DARPA under Contract No. HR00112190135 and HR00112290115 and NSF grants 1845216 and 2339898.

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

# A   APPENDIX

## A.1   MEASURING THE RANK OF POSSIBLE SOLUTIONS IN NOLA VS PRANC:

Choosing $n$ basis vectors ($d$ dimensional each) in PRANC will result in all possible learned matrices living in a $n$ dimensional subspace. However, since NOLA with the same number of total parameters ($k + l = n$) uses $A \times B$ factorization, the possible solutions can live in a higher dimensional subspace. We do a simple experiment by sampling several random coefficient vectors, reconstructing the $\Delta W$ matrix, reshaping it to be a long ($d^2$)-dimensional vector, and measuring the rank of the covariance of samples to see how much of the whole space is covered by the samples. The results are shown in Figure 2 for a simple experiment with varying $d$ and $n$. As expected, NOLA can cover the whole space (full-rank) using a small number of parameters compared to PRANC. Note that the rank in this analysis is on the covariance of possible random samples of weight matrices and should not be confused with the rank in LoRA or NOLA formulation.

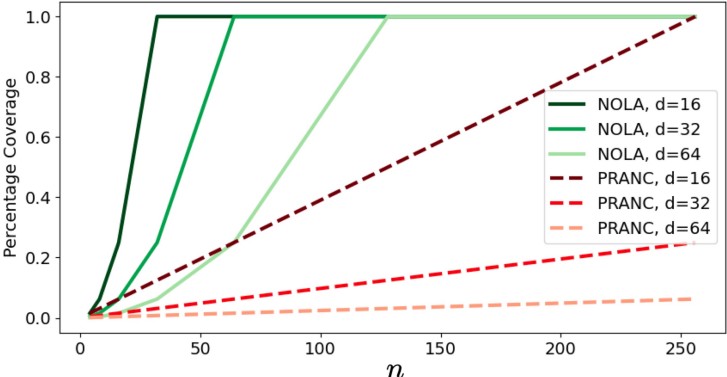

Figure 2: Comparing the rank of samples in the solution subspace for PRANC and NOLA, given the same number of parameters, $n$. "Percentage Coverage" is the subspace rank divided by the max possible rank ($d^2$), so $1.0$ denotes full rank. As expected, the coverage for PRANC increases linearly while it saturates very fast for NOLA.

## A.2   NOLA IN TRAINING FROM SCRATCH:

**MLP on MNIST (a toy experiment):**

We believe NOLA is a better reparametrization than PRANC and can achieve local minimums that PRANC cannot achieve. To show this empirically, we perform a very simple experiment where we apply a 2-layer MLP for the MNIST classification task. Since we want to measure which method is better at reaching the local minimums and not necessarily the generalization, we evaluate the models by the training loss. Also, we intentionally use a large number of neurons in the hidden layer to over-parameterize and increase the number of local minimums.

We use a linear layer from 784 features to 256 with bias followed by ReLU and a linear layer from 256 to 10 classes. We use all $60K$ samples for training to report the training loss. For both PRANC and NOLA, we use 32 parameters for each layer. Other hyperparameters are same for both: 200 epochs, 512 batch size with $lr = 0.05$. We use rank $r = 4$ for NOLA. PRANC has a final training loss of $0.87$, while NOLA achieves a training loss of $0.71$. This simple experiment empirically supports that NOLA has better representation power. Additionally, PRANC training finishes in 1152 seconds while NOLA finishes in 579 seconds. We will leave the theoretical study of this comparison for future work. Moreover, this experiment empirically shows that NOLA is a generic method, and its success is not dependent on the architecture of transformers or attention modules.

**CNN on ImageNet100 and CIFAR-10:**

Moreover, to compare the representation power of NOLA and PRANC, we train NOLA from scratch on an image classification task using a CNN architecture. For each convolution layer, we reshape

all parameters of a layer into a matrix (close to square shape) and apply NOLA to the matrix. Then, we reshape it to the original shape of the convolution. Additionally, we train LoRA using a similar approach as NOLA. We follow a similar setup as (Nooralinejad et al., 2022) for our experiments on image classification.

**Datasets and Architectures:** We consider two architectures in our experiments: ResNet20 with $270K$ parameters, and ResNet18 (He et al., 2016) with $11M$ parameters. We train ResNet20 on CIFAR10 (Krizhevsky et al., 2014), and ResNet18 on ImageNet100 (Deng et al., 2009).

**Results:** We report result of ImageNet100 in Table 9, and CIFAR10 in Table 8. NOLA outperforms both PRANC and LoRA with a similar number of parameters.

**Implementation Details:** For ImageNet100 and ResNet18, we use $k = l = 2,000$ basis for each of 20 modules, and for the classifier (last linear layer), we used $k = l = 10,000$, resulting in a total of $100,000$ trainable parameters excluding $9,600$ batchnorm parameters. We use rank 64 for all layers. We train all models using Adam optimizer with a learning rate of $0.001$ and batch size of 256 for 200 epochs. For CIFAR-10 and ResNet20, we use $k = l = 250$ basis for each convolutional module, and for the linear layer, we use $k = l = 1000$ parameters. We use batch size 256, Adam optimizer, and a learning rate of $0.001$. We use a single NVIDIA-GeForce RTX 3090 for all experiments.

**Training Time Comparison:** We measure the training time of NOLA and PRANC on a single NVIDIA-GeForce RTX 3090 GPU and batch size of 256. Note that training time includes both forward and backward passes for each batch. On average, NOLA processes a batch in 228ms while PRANC does the same in 1070ms, so NOLA is $4.6$ times faster than PRANC.

Table 8: **Training On CIFAR10:** Result of our method on CIFAR10 dataset and ResNet20.

| Method | # Params | Acc. |
|---|---|---|
| trained model | 269,722 | **88.92%** |
| PRANC | 12,752 | 81.5% |
| LoRA | 13,295 | 81.5% |
| NOLA | 12,876 | 82.4% |

Table 9: **Training On ImageNet100:** Result of our method on ImageNet-100 dataset and ResNet18

| Method | # Params | Acc. |
|---|---|---|
| trained model | 11,227,812 | **82.1%** |
| HashedNet (Chen et al., 2015) | 129,200 | 52.96% |
| PRANC | 119,200 | 61.08% |
| LoRA | 150,000 | 63.50% |
| NOLA | 109,600 | 64.66% |

## A.3 ABLATION AND DETAILS OF NOLA ON VISION TRANSFORMERS:

**Implementation detail:** We consider learning rates of $5e-3$, $1e-3$ and $5e-4$ for LoRA, NOLA and Linear methods and $8e-5$, $5e-5$, $3e-5$ and $1e-5$ for Full-FT. The best settings is chosen based on the performance on validation set. For creation of $k$-shot dataset, we randomly sample without replacement from the train set. For each of these sets, we run with three different initializations of the networks. This process is repeated four times and the averaged values are reported.

**Comparison between NOLA-QV and NOLA-MLP:** We experiment with NOLA layer in both the attention and MLP modules of the vision transformer. We observe that applying NOLA on MLP performs better than that on attention block (Table 10). Thus, we use NOLA-MLP as our default setting. Note that the number of trainable parameters remains the same in both versions. Unlike this, applying LoRA on MLP block would require significantly higher number of trainable parameters due to the increased dimensions of the weight matrices in MLP compared to those in attention block.

Table 10: **Comparison between NOLA in MLP and attention blocks:** We observe that NOLA on MLP block is more effective. We choose this as our default setting.

| Base Model | | # Train Params | CIFAR-10 | | CIFAR-100 | | CUB-200-2011 | | Caltech-101 | |
|---|---|---|---|---|---|---|---|---|---|---|
| | | | 5 | 10 | 5 | 10 | 5 | 10 | 5 | 10 |
| ViT-L | NOLA-QV | 47K | 87.0 (0.9) | 91.6 (0.7) | 74.8 (0.6) | 80.4 (0.9) | 75.3 (0.4) | 81.7 (0.3) | 87.9 (1.1) | 90.6 (0.5) |
| | NOLA-MLP | 47K | 87.9 (1.3) | 92.2 (0.5) | 75.1 (0.6) | 81.3 (0.8) | 75.5 (0.6) | 81.7 (0.4) | 88.0 (1.2) | 90.6 (0.5) |

## A.4 RESULTS OF NLG TASK ON DART AND WEBNLG DATASETS:

In Table 11, we report more results similar to Table 1 using GPT-2 M and GPT-2 L on DART (Nan et al., 2020) and WebNLG (Gardent et al., 2017) datasets.

Table 11: **DART and WebNLG Dataset**: Similar to Table 1 we compare NOLA to other methods. NOLA is on par or better with other methods with the same number of parameters.

| Method | Adapted Layers | Adapter Rank | # Trainable Parameters | DART BLEU↑ | MET↑ | TER↓ | WebNLG BLEU↑ | MET↑ | TER↓ |
|---|---|---|---|---|---|---|---|---|---|
| **GPT-2 M** | | | | | | | | | |
| Finetune | All Layers | - | 354.000M | 46.2 | 0.39 | 0.46 | 46.5 | 0.38 | 0.53 |
| Adapter[L] | Extra Layers | - | 0.370M | 42.4 | 0.36 | 0.48 | 50.2 | 0.38 | 0.43 |
| Adapter[L] | Extra Layers | - | 11.000M | 45.2 | 0.38 | 0.46 | 54.9 | 0.41 | 0.39 |
| Finetune[Top2] | Last 2 Layers | - | 24.000M | 41.0 | 0.34 | 0.56 | 36.0 | 0.31 | 0.72 |
| PreLayer | Extra Tokens | - | 0.350M | 46.4 | 0.38 | 0.46 | 55.1 | 0.41 | 0.40 |
| LoRA | QV | 4 | 0.350M | 47.1 | 0.39 | 0.46 | 54.9 | 0.41 | 0.39 |
| LoRA | QV | 1 | 0.098M | 46.4 | 0.38 | 0.48 | 53.5 | 0.40 | 0.40 |
| NOLA (Ours) | QV | 8 | 0.096M | 47.0 | 0.38 | 0.48 | 53.9 | 0.40 | 0.40 |
| NOLA (Ours) | MLP | 8 | 0.096M | 47.1 | 0.38 | 0.47 | 54.7 | 0.41 | 0.40 |
| NOLA (Ours) | QV | 8 | 0.048M | 45.7 | 0.38 | 0.49 | 53.8 | 0.40 | 0.40 |
| NOLA (Ours) | MLP | 8 | 0.048M | 45.5 | 0.38 | 0.49 | 53.0 | 0.40 | 0.40 |
| **GPT-2 L** | | | | | | | | | |
| Finetune | All Layers | - | 774.000M | 47.0 | 0.39 | 0.46 | 55.5 | 0.42 | 0.42 |
| Adapter[L] | Extra Layers | - | 0.880M | 45.7 | 0.38 | 0.46 | 56.0 | 0.41 | 0.39 |
| Adapter[L] | Extra Layers | - | 230.000M | 47.1 | 0.39 | 0.45 | 57.7 | 0.43 | 0.39 |
| PreLayer | Extra Tokens | - | 0.770M | 46.7 | 0.38 | 0.45 | 56.3 | 0.42 | 0.40 |
| LoRA | QV | 4 | 0.770M | 47.5 | 0.39 | 0.45 | 57.1 | 0.43 | 0.38 |
| LoRA | QV | 1 | 0.184M | 47.7 | 0.39 | 0.47 | 55.9 | 0.42 | 0.39 |
| NOLA (Ours) | QV | 8 | 0.144M | 47.8 | 0.39 | 0.47 | 55.8 | 0.41 | 0.39 |
| NOLA (Ours) | MLP | 8 | 0.144M | 47.8 | 0.39 | 0.47 | 56.0 | 0.42 | 0.39 |
| NOLA (Ours) | QV | 8 | 0.072M | 46.4 | 0.38 | 0.48 | 55.5 | 0.41 | 0.38 |
| NOLA (Ours) | MLP | 8 | 0.072M | 46.8 | 0.38 | 0.48 | 55.8 | 0.41 | 0.39 |

