# OpenReview forum: "NOLA: Compressing LoRA using Linear Combination of Random Basis"
_ICLR.cc/2024/Conference — ICLR 2024 poster_

### Official Review · Reviewer_CrTj · 2023-10-27

**Soundness:** 3 good
**Presentation:** 3 good
**Contribution:** 3 good
**Rating:** 6
**Confidence:** 4

**Summary:**

This paper proposes a technique to reduce the number of trainable parameters while fine-tuning large language models and large vision encoders. Their technique involves modeling the weight update matrix as a linear combination of fixed random matrices that are also rank-constrained. The fine-tuning process then involves learning just the coefficients in the linear combination. When the proposed technique is used, any updates to the models requires the communication/ storage of only the coefficients and the random seeds required to generate the codebook matrices (apart from the base weights of the original model of course).

The authors demonstrate that the proposed technique preserves the performance on the fine-tuning task while achieving large ( up to 1/20 x baseline models) reduction in the number of trainable parameters. This is shown in both language and vision domains. In the language domain, they show it on the NLG challenge dataset. In the vision domain, they show is on CIFAR, CUB and Caltech-101 datasets.

**Strengths:**

- The paper is well motivated and clearly written.
- Parameter-efficient fine-tuning is a popular area of research currently and this paper makes a good contribution to this area.
- The proposed technique is interesting and the results demonstrate that the method preserves performance while achieving a low parameter count
- The proposed method helps overcome the some of the limitations of methods such as LoRA, as described in the paper

**Weaknesses:**

I appreciate the results provided in the paper. But I think that some more in-depth evaluation and some more explanation of the current results would add value to the paper. I outline some specifics below.

Language experiments:
- Why does full fine-tuning achieve much lower performance in Table 1?
- Could the authors also provide the performance of the GPT-M, L models *without any fine-tuning* on the tasks considered? This will give the readers an idea of how much improvement is being achieved.

Vision experiments: I find the experiments provided in this section (3.3) a bit weak. The main reason being that the ViT models pre-trianed on imagenet are already pretty powerful. Fine-tuning these models on much smaller and easier datasets such as CIFAR may not be the best way to demonstrate usefulness. In particular, I have the following comments:

1. Can the authors try their techniques on more challenging datasets?
2. Although the number of parameters for the linear layer baseline depends on the dataset, it would be good to have this information visible in the Table.
3. As before, can the authors provide the performance of the models considered without any fine-tuning? (0-shot classification on the downstream datasets)

**Questions:**

1. What is the distribution used to generate the random basis matrices? Did the authors experiments with a few different choices?

---

> ### Author Response · Authors · 2023-11-23
> **Response to Reviewer CrTj**
>
> Thanks for your valuable feedback.
>
> **Addressing Weaknesses:**
>
> **Language experiments:**
>
> - Yes, full finetuning is worse which is consistent with the results in LoRA paper. We believe it is due to overfitting.
>
> - This is an interesting suggestion. we looked at the literature and could not find any prior work reporting zero-shot results for this E2E NLG benchmark. We tried it ourselves and got very low accuracy. We believe we need to engineer a prompt for this task since a non-finetuned model does not know the task. We will do this for the camera ready version.
>
> Meanwhile, we found the following paper [b] that shows zero-shot results on a related dataset, DART. They too report very low accuracy for the zero-shot setting: Their BLEU score is 13.26 for zero-shot and 33.41 for 3-shot. We also report the results for DART in the appendix, but our results are not directly comparable with theirs since our experiments are with GPT-M and GPT-L while they use GPT-XL.
>
> [b] Keymanesh, Moniba, Adrian Benton, and Mark Dredze. "What makes data-to-text generation hard for pretrained language models?." arXiv preprint arXiv:2205.11505 (2022).
>
> **Vision experiments:**
>
> 1. We agree and thank the reviewer for the suggestion. We perform experiments with NOLA and LoRA on various more challenging datasets. The results are reported in the following table and also in Table 6 of the revised paper. These are more challenging settings given an ImageNet pretrained model since the accuracy for some cases is around $30$% to $40$%. Interestingly, NOLA is on-par or better than LoRA in all cases with almost 3 times smaller number of parameters, so our conclusion holds on these more challenging datasets too.
>
> **5 shot:**
>
> | Method   | \# Train | Aircraft 5 Shot | Food101 5 Shot | Pets 5 Shot | SUN397 5 Shot |
> |----------|----------|------------------|-----------------|--------------|----------------|
> | NN       |          | 24.6             | 48.4            | 82.3         | 44.4           |
> | Linear   |          | 29.7 (2.3)        | 53.2 (0.3)      | 88.4 (0.4)   | 38.0 (0.8)      |
> | Full-FT  | 82M      | 31.4 (2.4)        | 48.6 (5.1)      | 91.1 (0.5)   | 45.0 (3.3)      |
> | LoRA     | 0.141M   | 32.4 (1.4)        | 60.8 (1.6)      | 91.6 (0.5)   | 51.6 (0.4)      |
> | NOLA-MLP | 0.047M   | 33.7 (2.2)        | 64.5 (0.8)      | 88.0 (0.6)   | 50.5 (0.4)      |
>
> **10 shot:**
>
> | Method   | \# Train | Aircraft 10 Shot | Food101 10 Shot | Pets 10 Shot | SUN397 10 Shot |
> |----------|----------|------------------|-----------------|--------------|----------------|
> | NN       |          | 27.1             | 54.2            | 86.2         | 51.5           |
> | Linear   |          | 36.9 (2.1)        | 61.5 (0.1)      | 91.3 (0.3)   | 42.7 (0.4)      |
> | Full-FT  | 82M      | 43.2 (2.0)        | 65.8 (2.7)      | 91.1 (0.5)   | 52.6 (0.3)      |
> | LoRA     | 0.141M   | 43.8 (1.5)        | 73.1 (0.6)      | 91.6 (0.5)   | 55.6 (0.3)      |
> | NOLA-MLP | 0.047M   | 43.3 (1.4)        | 72.6 (0.4)      | 92.2 (0.3)   | 55.5 (0.3)      |
>
>
> 2. We report the information on number of parameters of the linear layer here and in the caption of Table 5 of the revised version. The size is: 0.01M, 0.1M, 0.2M, 0.1M parameters for CIFAR-10, CIFAR-100, CUB and Caltech respectively. Similarly, the linear classifier sizes for the additional datasets considered in the revised submission are: 0.03M, 0.2M, 0.1M, 0.3M for Aircraft, Food101, Pets and SUN397 datasets respectively. We did not previously include this in the table since the linear layer classifier is constant for all methods and makes the table cluttered.
>
>
>
> 3. **Continued in Next Comment...**

---

> > ### Author Response · Authors · 2023-11-23
> > **Response to Reviewer CrTj (Continued)**
> >
> > 3. We consider nearest-neighbor (1-NN) evaluation for zero-shot classification on downstream tasks using ImageNet pretrained features. Note that zero-shot here indicates that the backbone feature extractor is not trained on any data from the downstream task. However, the train set of the downstream task is used as the train set in the nearest neighbor search. As expected, the finetuning methods (NOLA and Linear) are better than the zero-shot in accuracy.
> >
> > **5-shot**
> >
> > | Model   | Method   |   | CIFAR-10 5-shot | CIFAR-100 5-shot | CUB-200-2011 5-shot | Caltech-101 5-shot |
> > |--------|---------|----------|------------|-------------|----------------|---------------|
> > | ViT-B  | NN      |          | 79.6       | 52.4        | 71.9           | 84.1          |
> > | ViT-B  | Linear  |          | 80.8 (1.1) | 58.9 (0.9)  | 72.7 (0.4)      | 85.8 (0.8)     |
> > | ViT-B  | NOLA-MLP|          | 87.9 (1.3) | 75.1 (0.6)  | 75.5 (0.6)      | 88.0 (1.2)     |
> > |      |       |        |          |           |              |             |
> > | ViT-B-MAE| NN      |          | 18.2 | 5.8 | 13.2 | 28.2 |
> > | ViT-B-MAE| Linear  |          | 27.4 (1.9) | 15.7 (0.7)  | 12.7 (0.3)  | 66.9 (1.1) |
> > | ViT-B-MAE| NOLA-MLP|          | 55.1 (2.6) | 42.1 (1.4)  | 35.8 (1.5)      |  88.0 (1.2)      |
> > |      |       |        |          |           |              |             |
> > | ViT-L  | NN      |          | 88.7       | 68.9        | 77.4           | 88.4          |
> > | ViT-L  | Linear  |          | 84.1 (1.8) | 63.7 (1.3)  | 73.7 (0.6)      | 87.6 (0.9)     |
> > | ViT-L  | NOLA-MLP|          | 89.0 (3.6) | 83.6 (0.9)  | 80.8 (0.6)      | 90.0 (0.7)     |
> > |      |       |        |          |           |              |             |
> > | ViT-L-MAE| NN      |          | 33.5       | 15.2        | 16.9           | 57.4          |
> > | ViT-L-MAE| Linear  |          | 40.2 (2.3) | 22.6 (0.9)  | 15.2 (0.3)      | 75.2 (0.5)     |
> > | ViT-L-MAE| NOLA-MLP|          | 70.6 (3.8) | 51.7 (1.1)  | 36.9 (5.6)      | 87.4 (0.4)     |
> > |      |       |        |          |           |              |             |
> >
> > **10-shot**
> >
> > | Model   | Method   |   | CIFAR-10 10-shot | CIFAR-100 10-shot | CUB-200-2011 10-shot | Caltech-101 10-shot |
> > |--------|---------|----------|-------------|--------------|-----------------|----------------|
> > | ViT-B  | NN      |          | 80.8        | 59.2         | 78.0            | 87.5           |
> > | ViT-B  | Linear  |          | 85.1 (1.0)  | 64.5 (0.7)   | 79.2 (0.2)      | 88.5 (0.4)      |
> > | ViT-B  | NOLA-MLP|          | 92.2 (0.5)  | 81.3 (0.8)   | 81.7 (0.4)      | 90.6 (0.5)      |
> > |    |     |      |         |           |             |            |
> > | ViT-B-MAE| NN      |          | 19.8       |     9.8     |   25.3         |     40.7      |
> > | ViT-B-MAE| Linear  |          | 34.5 (1.4) | 22.2 (0.2) | 18.4 (0.3)      | 76.9 (0.6) |
> > | ViT-B-MAE| NOLA-MLP|          |  72.1 (2.7) |  53.5 (1.0)  |   53.9 (0.6)   |   90.6 (0.5)    |
> > |    |     |      |         |           |             |            |
> > | ViT-L  | NN      |          | 89.9        | 74.0         | 82.3            | 90.1           |
> > | ViT-L  | Linear  |          | 88.4 (1.1)  | 70.6 (0.9)   | 79.2 (0.3)      | 89.9 (0.4)      |
> > | ViT-L  | NOLA-MLP|          | 96.0 (0.5)  | 87.8 (0.6)   | 85.2 (0.2)      | 91.7 (0.3)      |
> > |    |     |      |         |           |             |            |
> > | ViT-L-MAE| NN      |          | 39.2        | 21.9         | 29.2            | 67.6           |
> > | ViT-L-MAE| Linear  |          | 49.2 (2.6)  | 31.3 (0.5)   | 21.9 (0.4)      | 83.2 (0.6)      |
> > | ViT-L-MAE| NOLA-MLP|          | 86.0 (1.4)  | 63.8 (0.8)   | 61.6 (1.0)      | 90.9 (0.5)      |
> > |    |     |      |         |           |             |            |
> >
> > **Addressing Questions:**
> >
> > 1. We simply use i.i.d. uniform distribution. We did not experiment with any other option. Studying what distribution should or may work better is an interesting research problem that we leave for the future work.

---

### Official Review · Reviewer_WWfE · 2023-10-30

**Soundness:** 2 fair
**Presentation:** 3 good
**Contribution:** 3 good
**Rating:** 6
**Confidence:** 3

**Summary:**

The paper looked at the problem of memory requirements for Low-Rank Adaptation (LoRA) and proposed NOLA to break the rank one lower bound present in LoRA. The core concept behind NOLA is to reparameterize a neural network using a linear combination of pseudo-randomly generated weights.

Thanks to the authors for a more detailed explanation of the motivations for the paper and for some of the latest research supporting them. Therefore, I will increase the rating by 1 point.

**Strengths:**

1. The paper discusses related works in detail and clearly summarizes its own contributions.
2. The paper performs extensive experiments to compare NOLA and existing PEFT solutions.
3. NOLA decouples trainable parameters from the choice of rank and the network architecture.

**Weaknesses:**

1. The work may need more rationales upfront to motivate the problems (i.e. the rank one lower bound present in LoRA). Given that mainstream GPUs have tens of GB of memory, it is reasonable to reduce the memory requirements from tens of GB to tens of MB at the expense of model quality through LoRA, as this can indeed reduce resource consumption and greatly reduce LLM transition overhead during inference. However, I don't think it makes much sense to further reduce memory requirements to several MBs at the expense of model quality.

**Questions:**

If users want to use the trained model on different versions of GPUs or software, how to ensure the consistency of the trained model? In such a situation, the same seeds can not generate the same pseudo-random matrices.

---

> ### Author Response · Authors · 2023-11-23
> **Response to Reviewer WWfE**
>
> Thanks for your valuable feedback.
>
> **Addressing Weaknesses:**
>
> 1. This is a great question that we have addressed in subsection named "Why Fewer Parameters Matter?" in the Introduction of the original submission. At the time of submission, we were envisioning that there will be several finetuned LLMs that need to be stored and used for inference. Interestingly a version of this vision is already here with the introduction of "GPTs" by OpenAI a couple of weeks ago. Task-specific GPTs are very easy to generate by the users, so we expect to see a large collection of them soon in the market place. Hence, handling the storage of those models and more importantly transitioning between them at the inference time is an interesting challenge that we suggest to solve. We argue that if the number of parameters are very small (e.g., in NOLA), we can store the weights for several GPTs in the GPU memory itself reducing the latency due to transferring them from CPU memory to GPU as soon as a new query for a specific GPT arrives. Then, even a small reduction in the size of the model compared to LoRA can become important due to the high cost of GPU memory and growing number of GPTs.
>
> Moreover, note that the sizes are smaller in these GPT-2 experiments compared to more advanced LLMs. Implementing LoRA and NoLA on larger LLMs will result in much bigger number of parameters. Hence, any reduction in size compared to LoRA will be important. Note that LoRA has a lower bound on the number of parameters corresponding to $r=1$ while NOLA is flexible and can trade accuracy with the number of parameters independent of the rank or feature dimension.
>
>
> Interestingly, a very recent paper [a] published a couple of weeks ago on arXiv (following) suggests that handling many task-specific LLMs is an important challenge and introduces a computer architecture solution. We are addressing a similar challenge by reparameterization at the learning time.
>
> [a] Sheng, Ying, Shiyi Cao, Dacheng Li, Coleman Hooper, Nicholas Lee, Shuo Yang, Christopher Chou et al. "S-LoRA: Serving Thousands of Concurrent LoRA Adapters." arXiv preprint arXiv:2311.03285 (2023).
>
> **Addressing Questions:**
>
> We did try the pseudo random generator with the same seed on several GPUs that we had access to (2080Ti, Quadro-6000, Quadro-8000, V100, P100, RTX-6000-Ada, and 3090). In all cases, the generated random sequences were the same, suggesting that these GPUs use the same random generator. Note that one can always run the random generator on the software (Python code) rather than relying on the efficient hardware implementation. Hence, even if the generator at a new device does not match with the one used in training, we can ship the parameters of the generator and run it on the software instead.

---

### Official Review · Reviewer_ba8s · 2023-11-04

**Soundness:** 3 good
**Presentation:** 3 good
**Contribution:** 3 good
**Rating:** 6
**Confidence:** 3

**Summary:**

The paper proposes a new approach for fine tuning LLMs for downstream tasks. The key idea is to replace the low rank updates of LoRA with linear combinations of fixed random matrices for which only the coefficients need to be tuned and stored in memory which significantly reduces the storage cost. The authors present experiments in both language and vision tasks where their approach preserve the accuracy of LoRA while reducing the parameter count by half or more.

**Strengths:**

1. The authors propose a novel, intuitive, and principled approach to address the problem of task based fine tuning of transformer based models.

2. The proposed approach shows significant reduction in storage overhead without compromising on accuracy across a range of experiments in both language and vision tasks.

**Weaknesses:**

1. The technical novelty is relatively minor with the overall idea being a combination of prior works PRANC and NOLA. While this seems enough to provide empirical improvement, the approach itself is not that big of an innovation over prior works.

2. While the prior approach PRANC is directly modified by the authors in this work there are no direct comparisons with it in either the language or vision tasks used to evaluate the proposed approach. There is a comparison of training loss in Section 3.4 and a comparison of the rank of possible solutions of the two approaches in Section 3.5 but without a direct comparison of test accuracy it is unclear if this approach is indeed an improvement over the baseline that it directly modifies.

**Questions:**

1. Why is the training time of NoLA with shared random basis similar to that of LoRA when the training time of NOLA with a unique random basis is higher? Aren't the number of coefficients being trained, the same in both cases?

2. The ablation study at the end of Section 3.1 appears inconclusive. Is there any takeaway on the effect of varying the rank in NOLA?

3. In Section 3.2 if only $\alpha$ and $\beta$ are quantized while A and B are not then won't that be less memory efficient than quantization in LORA?

4. Please highlight the entries in Table 5 with the best performance for a given scenario. Currently there are too many entries, and it is too difficult to figure out which method is better for which case.

5. If each matrix in PRANC has size $d^2$ then why do we need multiple matrices to cover the rank of the original $\Delta W$ matrix (which also has size $d^2$)?

---

> ### Author Response · Authors · 2023-11-23
> **Response to Reviewer ba8s**
>
> Thanks for your valuable feedback.
>
> **Addressing Weaknesses:**
>
> 1. PRANC reparameterizes CNN models for visual recognition tasks while training from scratch. We use a similar reparameterization but with several key distinctions:
>
> (a) We introduce a similar reparameterization to both A and B matrices of LoRA method and show on-par results with much smaller number of parameters.
>
> (b) We learn the residual model instead of learning it from scratch as in PRANC.
>
> (c) Moreover, PRANC is computationally expensive and has a large memory footprint due to the large basis matrix. We resolve this issue in NOLA since the big matrix is factorized into two smaller matrices. In the appendix, we show NOLA outperforms PRANC on vision tasks from scratch too.
>
> (d) In addition, in Figure 2, we show that the factorization in NOLA offers much higher coverage of the subspace compared to PRANC with similar number of parameters.
>
> 2. In the Appendix of the original submission, we directly compare NOLA with PRANC on vision tasks (Table 8 for CIFAR-10 and Table 9 for ImageNet-100). NOLA outperforms PRANC with on-par or smaller number of parameters. We report this in Appendix due to space limitations. PRANC is not suitable for language tasks due to its high cost of computation and memory for large models. We present the comparison with PRANC on ImageNet-100 below. The `Trained model' in the table refers to training the entire ResNet-18 model from scratch.
>
> | **Method**           | **\# Params** | **Acc.**    |
> |-|-|-|
> | trained model        | 11,227,812    | **82.1\%**  |
> |                   |           |         |
> | HashedNet Chen et al. (2015) | 129,200 | 52.96\% |
> | PRANC                | 119,200       | 61.08\%     |
> | LoRA                 | 150,000       | 63.50\%     |
> | NOLA                 | 109,600       | 64.66\%     |
>
> **Addressing Questions:**
>
> 1. Yes, when using non-shared basis, the training is almost $7$% slower compared to using shared basis. In shared basis case, the same basis are used across all the layers. We believe the extra training time is due to the latency of the random matrix generator since in the non-shared case, we call it once for each layer at each iteration. Table 2 shows that shared basis version works as well as the non-shared one.
>
> 2. The goal of that ablation study and Table 3 is to show the effect of changing the rank while keeping the number of parameters constant. This is possible in NOLA since we decouple the number of parameters and the rank. We show that the results do not change significantly with varying rank suggesting that in this setting, the rank does not necessarily add to the expressiveness of the model. Note that such a study is not possible with LoRA since increasing rank increases the number of parameters too. Here and in the revised version, we also report a new section in the table that shows the results when we vary the number of parameters while keeping the rank constant. As expected, the model improves with more number of parameters.
>
> | Model & Method                 | # Trainable | Rank | BLEU  | NIST | MET   | ROUGE-L | CIDEr |
> |-|-|-|-|-|-|-|-|
> | GPT-2 M (NOLA QV)             | 96K         | 8    | 70.03 | 8.82 | 46.74 | 71.64   | 2.51  |
> |                               | 96K         | 4    | 69.69 | 8.76 | 46.56 | 71.44   | 2.51  |
> |        | 96K         | 2    | 70.47 | 8.86 | 46.71 | 71.79   | 2.53  |
> |   | 96K         | 1    | 69.09 | 8.78 | 45.87 | 70.15   | 2.45  |
> | |  |  |   |  |  |  |  |
> | GPT-2 M (NOLA QV)             | 96K         | 8    | 70.03 | 8.82 | 46.74 | 71.64   | 2.51  |
> |    | 48K         | 8    | 70.09 | 8.82 | 46.44 | 71.36   | 2.52  |
> |    | 24K         | 8    | 68.30 | 8.67 | 45.13 | 68.91   | 2.40  |
> |   | 12K         | 8    | 67.18 | 8.60 | 43.99 | 68.38   | 2.26  |
>
>
> 3. $A$ and $B$ are not quantized, but we do not need to keep them in the memory. In the inference for a layer, we can calculate $A$ and $B$ using quantized $\alpha$ and $\beta$ and discard $A$ and $B$ before going to the next layer. Then, there will be very small memory overhead for storing $A$ and $B$ for a single layer. Hence, the memory will be almost the same as LoRA with similar number of parameters. Moreover, if the original model is not quantized, the memory savings with quantized LoRA during inference is minimal and may not matter.
>
> 4. Thanks for the suggestion. Note that simply highlighting the best is not enough since we are not claiming that NOLA is consistently better than LoRA in terms of accuracy. We want to show that they are on-par while NOLA is smaller in size. Hence, in the revised Table 5, we "boldface" the best number and "underline" any method within one point accuracy difference of the best method to show that NOLA and LoRA are close. Hope this helps.
>
> 5. Since PRANC matrices are initialized with random values and are not optimized, we need to combine several of them linearly (with learned coefficients) to represent $\Delta W$.

---

> > ### Comment · Reviewer_ba8s · 2023-11-23
> > **Thank you for your response**
> >
> > Thank you for the responses. My concerns have been adequately addressed. As I had already recommended accepting the paper, I will keep my score unchanged.

---

### Official Review · Reviewer_P5nc · 2023-11-05

**Soundness:** 3 good
**Presentation:** 2 fair
**Contribution:** 3 good
**Rating:** 6
**Confidence:** 4

**Summary:**

Low Rank Adaptation (LoRA) presents a series of drawbacks, particularly its constrained parameter reduction due to rank-1 matrices, which cannot be further diminished. Additionally, LoRA's parameter count is heavily reliant on the model's architecture. In response to these limitations, this paper suggests an innovative solution by advocating the use of a linear combination of random projections to replace LoRA's update matrix, effectively addressing the issues mentioned earlier. This approach is inspired by the previous paper known as PRANC. Personally, I found the paper to be a valuable source of knowledge and a unique one, appreciating its quirky yet straightforward idea

**Strengths:**

**Leveraging Ideas from Other Papers for Enhanced Parameter Efficiency:** This paper skillfully incorporates concepts from existing research to optimize parameter efficiency.

**Achieving Comparable or Superior Performance to NOLA:** This research attains performance levels akin to LoRA while significantly enhancing parameter efficiency.

**Weaknesses:**

**Poorly presented results**-  The main issue in the presentation of the results lies in their lack of clarity and explanatory depth. Firstly, the results fail to offer any substantial insights into how the method operates, leaving readers without a clear understanding of the underlying mechanisms. Additionally, Tables 1 and 5 are presented as mere lists of numbers without the necessary context or explanation, making it challenging for the audience to derive meaningful conclusions from the data. A critical element that appears to be missing is a discussion of what works better and the reasons behind it, which is crucial for a comprehensive understanding of the findings. To improve the presentation of the main results, it is essential to provide better explanations and context for the data, as well as a deeper analysis of what drives the observed outcomes.

**Questions:**

1. GPT2-L and GPT-2M seems to perform the same for LoRA. Is there any explanation on why this is the case?
2. The presentation of results preceding the training details in Section 3.1 seems to be an inadvertent oversight. To enhance the logical flow of the content, it is advisable to reverse the order, placing the training details before the results.What happens when you increase the number of parameters for NOLA?  - Does it perform better than LoRA. For example results of NOLA with 0.35M parameters
3. How does NOLA's performance change when the number of parameters is increased? Does it outperform LoRA? For instance, are there any results available for NOLA with 0.35 million parameters?

---

> ### Author Response · Authors · 2023-11-23
> **Response to Reviewer P5nc**
>
> Thanks for your valuable feedback.
>
> We argue that there are many possible local minima for $A$ and $B$ matrices in LoRA method. Training LoRA results in finding one of those local minima. Inspired by PRANC, our main intuition is that one of those local minima may live in a low dimensional random subspace. Hence, in learning, we directly look for that local minimum instead of searching for the whole space. Since the subspace is low-dimensional, we save on the number of parameters. Our experiments support our intuition empirically as we do not loose accuracy compared to original LoRA while reducing the number of parameters and decoupling them from the rank in LoRA method. We will clarify this in the text.
>
> As stated, our main conclusion from the experiments is that our method can reduce the number of parameters while getting on-par results compared to LoRA. Since the numbers are on-par with LORA, we do not specifically highlight the best numbers on Table 1. We believe a part of confusion for the reviewer on Table 1 is due to showing that different variants (MLP, QV, etc) of our method achieves on-par results too. We will move some of those variants to the appendix or ablation section in the camera ready.
>
>
> **Addressing Questions:**
>
> 1. Yes, they are similar on LoRA paper too. One explanation may be that GPT-2 M has enough capacity for these tasks, but we are not sure about a clear explanation. This may need to be studied as future work.
>
> 2. That's a good idea. We changed the order in the revised version to make it more clear.
>
> 3. We perform this experiment where the number of parameters for NOLA and LoRA are the same (0.35M). The results are shown below and in Table 1 of the revised paper. Our results are again on-par with LoRA suggesting that one can use NOLA instead of LoRA even if the size of LoRA is suitable for the application. Note that we never claim that our method outperforms LoRA in language tasks. We simply state that our method is capable of reducing the size while the results are on-par with LoRA.
>
>
> | Model & Method        | # Trainable | BLEU | NIST | MET  | ROUGE-L | CIDEr |
> |-----------------------|-------------|------|------|------|---------|-------|
> | GPT-2 M (LoRA r=4)    | 0.350M      | 70.4 | 8.85 | 46.8 | 71.8    | 2.53  |
> | GPT-2 M (NOLA QV)     | 0.350M      | 70.1 | 8.80 | 46.8 | 71.7    | 2.53  |

---

> > ### Comment · Reviewer_P5nc · 2023-11-23
> > **Good work**
> >
> > Thank you for addressing the concerns. It has been addressed well. I will keep my scores unchanged.

---

### Author Response · Authors · 2023-11-23
**Response to All Reviewers and AC**

We thank the reviewers for their valuable feedback. All reviewers acknowledge that our ideas are interesting, timely, and effective. For instance: Reviewer P5nc calls our ideas quirky yet straightforward, and a good source of knowledge; Reviewer ba8s says it is a novel, intuitive and principled idea that results in significant reduction in storage overhead without compromising on accuracy; Reviewer WWfE acknowledges our extensive experiments and says our ideas decouples trainable parameters from the choice of rank and the network architecture; Reviewer CrTj calls it well-motivated, well-written and timely, making a good contribution to this area. We have addressed the concerns individually for each reviewer. We also provide an updated version of our main submission reflecting some of the requested changes highlighted in blue.

**Key idea:** We argue that there are many possible local minima for $A$ and $B$ matrices in LoRA method. Training LoRA results in finding one of those local minima. Inspired by PRANC, our main intuition is that one of those local minima may live in a low dimensional random subspace. Hence, in learning, we directly look for that local minimum instead of searching for the whole space. Since the subspace is low-dimensional, we save on the number of parameters. Our experiments support our intuition empirically as we do not loose accuracy compared to original LoRA while reducing the number of parameters and decoupling them from the rank in LoRA method.

---

### Meta-Review · Area_Chair_GYu6 · 2023-12-05

**Metareview:**

The paper proposes an alternative to LoRA fine-tuning of LLMs (and also ViTs) for downstream tasks. Based on a related approach, PRANC, the key idea is to replace the low rank updates of LoRA with linear combinations of fixed random matrices for which only the coefficients need to be tuned and stored in memory which significantly reduces the storage cost. The authors present experiments in both language and vision tasks where their approach preserves the accuracy of LoRA while reducing the parameter count by half or more.

**Justification For Why Not Higher Score:**

Limited novelty due to similarity to existing similar work (PRANC), explanation/motivation of why this approach works well is not easily understandable

**Justification For Why Not Lower Score:**

The paper provides comprehensive experiments, the author response addressed the reviewers concerns

---

### Decision · Program_Chairs · 2024-01-16

Accept (poster)